# Behavioral Alterations and Decreased Number of Parvalbumin-Positive Interneurons in Wistar Rats after Maternal Immune Activation by Lipopolysaccharide: Sex Matters

**DOI:** 10.3390/ijms22063274

**Published:** 2021-03-23

**Authors:** Iveta Vojtechova, Kristyna Maleninska, Viera Kutna, Ondrej Klovrza, Klara Tuckova, Tomas Petrasek, Ales Stuchlik

**Affiliations:** 1National Institute of Mental Health, Topolova 748, 25067 Klecany, Czech Republic; Kristyna.Maleninska@nudz.cz (K.M.); Viera.Kutna@nudz.cz (V.K.); o.klovrza@gmail.com (O.K.); Klara.Tuckova@nudz.cz (K.T.); Tomas.Petrasek@nudz.cz (T.P.); 2Laboratory of the Neurophysiology of the Memory, Institute of Physiology of the Czech Academy of Sciences, Videnska 1083, 14220 Prague 4, Czech Republic; 3First Faculty of Medicine, Charles University, Katerinska 32, 12108 Prague 2, Czech Republic; 4Faculty of Science, Charles University, Albertov 6, 12800 Prague 2, Czech Republic

**Keywords:** lipopolysaccharide, maternal immune activation, prenatal infection, chronic bacterial infection, parvalbumin-positive interneurons, macrocephaly, schizophrenia, autism, sex differences, development

## Abstract

Maternal immune activation (MIA) during pregnancy represents an important environmental factor in the etiology of schizophrenia and autism spectrum disorders (ASD). Our goal was to investigate the impacts of MIA on the brain and behavior of adolescent and adult offspring, as a rat model of these neurodevelopmental disorders. We injected bacterial lipopolysaccharide (LPS, 1 mg/kg) to pregnant Wistar dams from gestational day 7, every other day, up to delivery. Behavior of the offspring was examined in a comprehensive battery of tasks at postnatal days P45 and P90. Several brain parameters were analyzed at P28. The results showed that prenatal immune activation caused social and communication impairments in the adult offspring of both sexes; males were affected already in adolescence. MIA also caused prepulse inhibition deficit in females and increased the startle reaction in males. Anxiety and hypolocomotion were apparent in LPS-affected males and females. In the 28-day-old LPS offspring, we found enlargement of the brain and decreased numbers of parvalbumin-positive interneurons in the frontal cortex in both sexes. To conclude, our data indicate that sex of the offspring plays a crucial role in the development of the MIA-induced behavioral alterations, whereas changes in the brain apparent in young animals are sex-independent.

## 1. Introduction

Maternal immune activation (MIA) represents a severe environmental risk factor for the development of some neuropsychiatric disorders with unclear etiology, such as schizophrenia or autism spectrum disorders (ASD) [1,2]. Both schizophrenia and ASD are neurodevelopmental disorders [3,4,5] sharing several symptoms, such as abnormal thinking, deficits in social behavior, impaired communication, cognitive deficits, anxiety, or inflexibility in novel situations. Other symptoms seem to be disease-specific, e.g., hallucinations and delusions in schizophrenia, and childhood-onset, repetitive ritualized behavioral patterns, and poor eye contact in ASD [4,6,7]. Nevertheless, there is a huge heterogeneity in the manifestation of particular symptoms in individual patients, not to mention the “combined” or “intermediate” diagnoses, such as childhood (early-onset) schizophrenia or schizophrenic autism [8,9,10]. From a historical perspective, it is notable that the term “autism” itself was originally coined to describe the social withdrawal in schizophrenia [10].

On the level of brain morphology, typical “schizophrenic” and “autistic” patients show different characteristics. Schizophrenia usually shows a reduction of hippocampal volume, enlargement of brain ventricles, or reduced thickness of the cortex [11]. In ASD, morphological findings are less consistent, but rather the enlargement of brain volume, particularly of the cerebellum, and various alterations of the cortical thickness or hippocampal volume have been reported [12,13,14,15,16].

Recent studies also suggest that deficits of parvalbumin-positive (PV+) interneurons, leading to disrupted inhibition of neuronal networks, could be one of the mechanisms behind abnormal thinking and psychosis in schizophrenia [17,18,19,20]. The decreased number or function of PV+ interneurons could have a fatal impact on correct neuronal signaling and the balance of excitation/inhibition leading to the discoordination of neuronal networks (reviewed in [21]). However, genetic manipulations in mice showed the connection of PV+ interneurons not only to schizophrenia-like [22,23] but also ASD-like phenotype [24,25,26].

In laboratory rodents, MIA can be modeled by application of viral particles, or more often their synthetic RNA analog polyinosinic:polycytidylic acid (polyI:C), and bacterial lipopolysaccharide (LPS) during a critical period of early development [27,28,29,30,31,32]. LPS is a major component of the cell wall of Gram-negative bacteria, such as *Escherichia coli* or *Salmonella*, causing strong immune system reaction with the release of pro-inflammatory cytokines, such as interleukin (IL)-6, IL-1β or tumor necrosis factor (TNF)-α [33,34,35,36]. LPS administered to pregnant dams caused an increase of pro-inflammatory cytokines in the fetal rat brain even after a single injection [37]; higher cytokine levels after chronic prenatal exposure to LPS could persist in adulthood [38].

It has been shown that MIA induced by viral or bacterial infection during pregnancy alters brain development and causes behaviors resembling schizophrenia or ASD in the rodent offspring (see [39] for a review). However, research in MIA models is laden with discrepancies in methodology between studies. Therefore, it is extremely difficult to generalize and judge the validity of the MIA models. In addition, clinical diagnostics in humans is based predominantly on verbally expressed symptoms whereas only objective signs (behavioral or neurobiological) are available in rodent models. This makes the difference between ASD-like and schizophrenia-like phenotypes in animals even more indistinct and subjective.

The aim of this study was to investigate a behavioral phenotype and brain characteristics of Wistar rats prenatally exposed to bacterial lipopolysaccharide (from *E. coli*, subcutaneous injections of a dose 1 mg/kg, from 7th day of a pregnancy to the delivery, every other day), and to validate this chronic MIA model with regard to schizophrenia and ASD. The relevant domains of behavior in the LPS- and saline-exposed offspring of both sexes were tested twice, in early adolescence and adulthood, to provide insights into the ontogeny of behavioral signs, and to evaluate the face validity of the model. We hypothesized that sex would also affect the character or time course of the behavioral alterations, because in both schizophrenia and autism, there are sex-dependent differences in prevalence and severity [40,41,42]. To evaluate the construct validity, we examined the frontal cortex and dorsal hippocampus of the juvenile offspring for PV+ interneuron numbers, and the overall morphology of the key structures in search of early endophenotypes. We hypothesized that the LPS juveniles would show a decreased number of PV+ interneurons and morphological changes corresponding to schizophrenia- or ASD-like phenotypes. We also measured sex effects and behavioral developmental changes in our study.

## 2. Results

### 2.1. Brain Analysis in P28-Old Rats

#### 2.1.1. Macrocephaly without Changes in Other Brain Morphology Parameters in Rats after Prenatal Exposure to LPS

Both rat males and females born to LPS-treated dams exhibited larger brain size, which was measured as the area of four different brain sections, compared to the offspring of saline-treated dams, as shown by the three-way analysis of variance with repeated measures (3wANOVA-RM), suggesting overall enlargement of the brain (Figure 1a, Table A2 in the Appendix C). Although the graph in Figure 1b indicates a thicker cortex in females prenatally exposed to LPS, two-way ANOVA (2wANOVA) did not show the difference to be significant (Table A2). The ventricle area (Figure 1c) as well as the dorsal hippocampal area (Figure 1d), measured relative to the whole brain area, were also not affected by prenatal LPS exposure, as 2wANOVA showed no significant difference between groups (Table A2).

*Sex differences*. In general, juvenile males had bigger brains than females, shown as larger areas of four different brain sections, as found by 3wANOVA-RM (Figure 1a, Table A2). We hypothesized that the observed differences in brain size could be driven by body size, but Pearson correlation between body size and brain slice areas showed no signs of significance (not shown). In addition, males had thicker cortex than females as found by 2wANOVA (Figure 1b, Table A2). As cortical thickness was not corrected for brain size, it is possible that the cortex was simply proportional to the larger brain in males. No sex differences were found in the ventricle or dorsal hippocampal area (Figure 1c,d, Table A2), where the relative area of the structure (compared to the slice area) was evaluated.

#### 2.1.2. Decreased Number of PV+ Interneurons, but No Changes in Microglia Number, in Rats after Prenatal Exposure to LPS

In animals prenatally treated by LPS, 2wANOVA showed a decreased number of PV+ interneurons in the upper part of the frontal cortex (area 1) compared to controls. The difference was found in both sexes and was more pronounced in the left hemisphere than in the right hemisphere as 2wANOVA analyses showed (Figure 2a, Table A3). In the lower part of the frontal cortex (area 2), the difference between groups in the number of PV+ interneurons was not significant (Figure 2b, Table A3). In 2wANOVA analyses, no effect of LPS was apparent in the dorsal hippocampus (Figure 2c, Table A3). Regarding the microglia, 2wANOVA did not find any significant difference between groups in the number of microglia in the dentate gyrus (DG) of the dorsal hippocampus (Figure 2d, Table A3).

*Sex differences*. Males had higher number of PV+ interneurons in the upper part of the frontal cortex (area 1) than age-matched females, specifically in the left hemisphere (Figure 2a, Table A3), but lower PV+ interneurons number in the dorsal hippocampus with a greater difference in the CA1-3 areas than in the dentate gyrus, all found by 2wANOVA analyses (Figure 2c, Table A3). The numbers of PV+ interneurons in the lower part of the frontal cortex (area 2; Figure 2b), or microglia in the dorsal hippocampus (Figure 2d), were not significantly different within sex (Table A3).

### 2.2. Behavior Analysis in P45- and P90-Old Rats

#### 2.2.1. Social Behavior and Communication Deficits after Prenatal LPS Exposure Manifest Earlier in Males Than in Females

Social behavior and communication showed changes caused by prenatal experience with LPS; however, LPS had a distinct effect on males and females. Regarding duration of non-anogenital social contact, 3wANOVA-RM revealed a significant interaction of group*sex, showing that LPS males spent less time by social contact than control males at both P45 and P90, while female groups did not significantly differ at any age (Figure 3a, Table A4). In the duration of anogenital exploration, no effect of LPS was found by 3wANOVA-RM (Figure 3b, Table A4).

The changes caused by prenatal LPS were also apparent in ultrasonic vocalization (USV) parameters. In general, 3wANOVA-RM did not show a significant effect of LPS on the total number of calls (Table A5). Nevertheless, 2wANOVA analyses done separately for P45 and P90 revealed group*sex interaction in adolescents showing that LPS males vocalized less and LPS females vocalized more in comparison to controls, which indicates the opposite effect of LPS on males and females at P45 (Figure 4a, Table A5). No significant effect of prenatal LPS specifically on trill-like USV was found by 3wANOVA-RM (Figure 4b, Table A5).

3wANOVA-RM did not reveal any significant difference between groups in the average duration of a simple call (Figure 4c, Table A5); however, it showed a general effect of LPS on the average duration of a composite call (Figure 4d, Table A5). Further analysis by 2wANOVA revealed that the difference is apparent only in adult females and not in males, as shown by interaction group*sex, as P90-old females of LPS-treated dams emitted shorted composite calls on average compared to females of saline-treated dams.

The ratio of high-frequency simple calls to composite calls duration in each group was calculated, and 3wANOVA-RM found a significant overall effect of prenatal exposure to LPS on this parameter. Separate analyses by 2wANOVA revealed a more pronounced effect in adulthood when the ratio of composite calls was decreased in the LPS rats compared to controls (Figure 4e, Table A5). This may indicate a decreased complexity of vocalization patterns in the LPS rats. In adolescent rats, the effect of LPS was not confirmed by a statistical analysis of 2wANOVA, but the graphs in Figure 4e indicate the same trend in P45-old LPS males, while P45-old LPS females did not differ from controls (Table A5).

*Sex differences*. In the social interaction test, 3wANOVA-RM found an interaction of age*sex showing that the total time of non-anogenital social contacts was higher in males than females at P45 (shown by 2wANOVA, although a simple effect did not show the effect), but not in adulthood (Figure 3a, Table A4). Duration of anogenital exploration did not differ between males and females (Figure 3b, Table A4). In addition, males emitted a higher total number of calls than females (Figure 4a) and a higher number of trill-like elements (Figure 4b), as 3wANOVA-RM analyses revealed (Table A5). In addition, the average duration of a simple call (Figure 4c), as well as a composite call (Figure 4d), was longer in males than females, according to 3wANOVA-RM analyses (Table A5). The proportion of simple versus composite calls was lower in P45 males, compared to P45-old females shown by 2wANOVA (Figure 4e, Table A5). This shows that males vocalized more in general and their ultrasonic utterances were both longer in duration and more complex in structure. This agrees with the observations of Potasiewicz et al., who found more pronounced playful behavior in P31–P32 males, which was accompanied by both quantitatively and qualitatively richer ultrasonic vocalization production relative to age-matched females [43]. They hypothesized that playful behavior serves as a training for behaviors establishing adult social hierarchy (dominance and submission displays, fights etc.), which is more important for rat males than females. Our P45 rats did not show playful behavior (which could be explained by both their age and the experimental settings), but it is probable that they transferred their richer vocal repertoire to non-playful social encounters as well.

*Developmental changes.* In the social interaction test, adult animals explored the anogenital area of their social partners more than P45 adolescents, as found by 3wANOVA-RM analysis (Figure 3b, Table A4). The duration of non-anogenital social contact did not significantly change during development (Figure 3a, Table A4). Anogenital exploration has an important role in individual recognition and determining the social and reproductive status of a social partner. It is possible that adult rats were more motivated to assess this type of information than adolescents. In total, adult rats emitted much more calls than adolescents (Figure 4a) and also more trill-like elements (Figure 4b), as shown by 3wANOVA-RM (Table A5). Moreover, the average duration of a simple call was longer in adults than in adolescent rats (Figure 4c, Table A5); this remained only a trend in case of a composite call (Figure 4d, Table A5). It seems that ultrasonic vocalization grows more complex in rat ontogeny, from simple neonatal calls to the full repertoire observed in mature individuals. Our results show that P45 rats still do not exhibit a fully mature vocal communication pattern.

#### 2.2.2. Increased Anxiety in Rats after Prenatal Exposure to LPS

The offspring of LPS-treated dams traversed the beam in the beam walking test (BW) faster than the controls, on the wide beam, as well as the narrow beam, independently on sex or age, as was found by 3wANOVA-RM analyses (Figure 5, Table A6). Visual observation indicated that higher anxiety motivated the LPS animals to leave the beam faster, since the control animals appeared calmer and took more time to explore their surroundings. The number of footslips in BW was not significantly changed by LPS experience, and the overall number of errors was low (Figure A2, Table A10 in the Appendix D). In the open field test (OF), 3wANOVA-RM revealed an interaction of group*age in the time spent in the arena center showing increased anxiety in the adult LPS offspring of both sexes, which spent more time near the walls of the OF compared to controls, as shown by simple effects (and confirmed by 2wANOVA). This difference was not present in adolescence (Figure 6a, Table A7). The anxiety-measuring parameters in the elevated plus maze test (EPM)—the proportion of open arm visits to total arm visits and frequency of risk assessment behavior—were not significantly changed by prenatal LPS exposure at any age of the offspring, as shown by 3wANOVA-RM analyses (Figure 6c,d, Table A8).

*Sex differences*. In BW, males needed more time to traverse the narrow beam than females, as analyzed by 3wANOVA-RM (Figure 5, Table A6). The number of footslips was not different between sexes (Figure A2, Table A10). It is probable that males were slower because they are larger and heavier than females. On the other hand, males seemed to be more anxious in the EPM, where they had lower open arm visits per total arm visits ratio, and they also had less risk assessment behavior than females, as shown by 3wANOVA-RM analyses (Figure 6c,d, Table A8). In the OF, no significant difference was found in the time spent in the center of the OF arena between males and females (Figure 6a, Table A7). It is probable that the observed effects were again driven by the sexual dimorphism in size. The open arms of EPM were relatively narrower for the larger males, explaining their avoidance by substantiated fear of falling. In the OF, where size of the animal did not matter so much, males did not show any signs of elevated anxiety. However, other task-specific effects related to motivation of the animals might also play a role.

#### 2.2.3. Decreased Activity in Rats after Prenatal Exposure to LPS

The analysis of walked distance in OF by 3wANOVA-RM revealed an interaction of age*group*sex, which means that prenatal exposure to LPS had a different effect on locomotion in adolescence and adulthood, depending on sex. At P45, LPS-affected males walked a similar distance as controls, while LPS-affected females were more active than control females (interaction group*sex found by 2wANOVA). At P90, LPS groups of both sexes showed slightly decreased locomotor activity in comparison to control rats (Figure 6b, Table A7), although 2wANOVA showed only a strong trend. In EPM, an interaction of group*age, demonstrated by 3wANOVA-RM, showed that the activity of LPS-exposed animals was not changed at P45, and it was slightly decreased at P90 in both sexes compared to saline-exposed rats (Figure 6e, Table A8). However, this difference was not corroborated by simple effects and was only shown as a trend by 2wANOVA.

*Sex differences*. Males walked a shorter distance in the OF arena than females (Figure 6b, Table A7). 3wANOVA-RM also found males to be less active in EPM, as measured by total arm visits (Figure 6e, Table A8). Again, the most straightforward explanation is body weight. In the EPM, anxiety could also contribute to the decreased exploratory activity.

*Developmental changes*. Adult animals walked much shorter distance in the OF arena compared to adolescent animals, as found by 3wANOVA-RM (Figure 6b, Table A7). Concerning anxiety, adults manifested a lower number of open arm visits per total arm visits, and they did less risk assessment behavior in comparison to adolescent animals, according to 3wANOVA-RM analyses (Figure 6c,d, Table A8). In the OF, there was no significant developmental alteration in the time spent in the middle of the arena (Figure 6a, Table A7). Behavior in OF and EPM can be explained by the body weight of the animals. Larger and heavier rats generally tend to be less active and avoid the narrow open arms due to risk of falling; this holds for males versus females as well as adults versus adolescents. Adult rats might also have been less motivated to explore, perhaps because they remembered the apparatus from the previous session. The analysis by 3wANOVA-RM revealed that adult rats needed less time to walk across the wide beam than adolescent rats (Figure 5, Table A6). This is rather surprising, given that adult rats were larger, heavier, and generally less active. However, the higher anxiety in adults could have played a role, motivating them to leave the beam as soon as possible. Moreover, they could have taken advantage of the previous experience with the task.

#### 2.2.4. Increased Startle Reaction in Adult Rats and PPI Deficit in Adult Females, but Not in Males, after Prenatal Exposure to LPS

The offspring born to LPS-treated dams manifested a higher startle response to a strong acoustic stimulus (120 dB) without prepulses compared to those from saline-treated dams, as shown by 2wANOVA; the effect was more pronounced in males (Figure 6f, Table A9). Moreover, 2wANOVA found a significant interaction of group*sex for startle reaction to strong acoustic stimulus 120 dB with 80-dB prepulse, as there was no effect in LPS-influenced males but a substantial decrease of prepulse inhibiton (PPI) in LPS females (also confirmed by *t*-tests done specifically in females, for all prepulses averaged, and especially for 80-dB prepulse), in comparison to controls (Figure 6g, Table A9). 3wANOVA-RM also showed a strong effect of prepulse loudness; the louder the prepulse, the higher inhibition of startle reaction the rats manifested (Table A9).

*Sex differences.* 2wANOVA showed that males demonstrated a higher startle reaction than females (Figure 6f, Table A9); however, this effect probably reflects the technical properties of the sensors, as the apparatus was more sensitive to startle-induced jumping in heavier animals.

## 3. Discussion

We have shown that prenatal exposure to MIA induced by LPS administration leads to changes in the brain at P28, and altered behavior at P45 and P90, resembling both schizophrenia and ASD. At the level of construct validity, measured brain parameters show similar alterations in males and females at juvenile age. However, at the level of face validity, the outcome is modulated by sex. In males, the behavioral changes appeared earlier, with social and communication deficits persisting from adolescence to adulthood, whereas in females, communication and PPI impairment only surfaced in adulthood.

We observed the numbers of γ-aminobutyric acid (GABA)ergic PV+ interneurons to be strongly decreased in the frontal cortex but unchanged in the dorsal hippocampus of P28 rats born to LPS-treated mothers (Figure 2). Comparable studies found decreased numbers of PV+ interneurons in adult rats after chronic LPS-induced MIA in both the frontal cortex and the hippocampus [44,45]. It seems that changes in GABAergic interneurons substantially contribute to the pathogenesis of schizophrenia. The patients exhibit impairment in GABAergic neurotransmission [46], as shown by a reduction of glutamate decarboxylase (GAD67), catalyzing the synthesis of GABA in interneurons [47], decreased levels of parvalbumin in cortical PV+ interneurons [48], and a decreased number of PV+ cells in the hippocampus [49]. Recently, it has been suggested that myelination of PV+ interneurons could be disrupted as well [20]. Interneuron dysfunction may lead to the observed impairment of cortical gamma oscillations and synchrony of neural networks [50,51]. Nevertheless, a decreased density of PV+ interneurons in the prefrontal cortex was also found in patients with ASD [18]. Mouse studies show that PV ablation or *N*-Methyl-d-aspartate (NMDA) receptor knockout in PV+ interneurons leads to ASD-like phenotype [25,26].

Our examination of juvenile brain morphology showed macrocephaly (Figure 1a) with unchanged size of lateral ventricles and dorsal hippocampus and cortical thickness. The number of microglia, at least within the dorsal hippocampus, was also not affected by prenatal LPS administration. Brain morphology after LPS MIA has been rarely described in the literature. Morphological changes of the brain were reported to be absent at P60 after a single prenatal LPS injection [52]. Increased microglial activation was found in mice and rats immediately after LPS injection [53,54]; however, little is known about the long-term effects of prenatal LPS exposure on microglia density or activation. Macrocephaly is also seen in a subpopulation of children with ASD, sometimes only at a specific age [13,55,56,57]. The size of lateral ventricles and hippocampus as well as cortical thickness are known to be affected in schizophrenia [11,58] and ASD [14,15,16], unlike our model. However, we examined very young animals, and it is possible that these changes would develop later.

Despite the sex-independent structural changes in juvenile brains, behavioral manifestations in later life differed between males and females. Decreased social contact and a reduced number and complexity of ultrasonic vocalizations appeared in LPS males already at P45. In females, comparable communication deficit manifested only at P90, although the duration of social contact itself was not changed (Figure 3 and Figure 4). Other studies investigating similar models also found the behavioral changes to be strongly modulated by sex [44,45]. Regarding early behavioral changes in animal models of MIA, the published evidence is ambiguous. Decreased social and play behavior of rats at P60 and P30, respectively [52], and reduced USV at P3 and P5 [27] after prenatal LPS treatment, were demonstrated. In contrast, no effect at P40 and P60 [33] or only subtle impairments in the social behavior of P70 rats [59] were reported elsewhere. However, it is necessary to note that there are considerable variations in MIA protocols used between different studies regarding species and strain, LPS serotype, route of administration, dosing, or timing, making direct comparisons difficult. A different class of models based on direct early postnatal applications of LPS to rat pups also manifest altered USV [54,60]. Social and communication deficits are typical symptoms of ASD that manifest early in life, and schizophrenia features disorganized, plain, or otherwise abnormal speech [61,62]. Our findings correspond to the situation in human patients. Contrary to social behavior, anxiety-like behaviors observed in our study were sex-independent. Shorter latencies in BW, as seen in both adolescents and adults (Figure 5), could be explained by higher anxiety, motivating the LPS rats to reach the homecage. This is supported by OF, where adult LPS rats showed more anxious behavior (Figure 6a). Better locomotor abilities of the LPS-exposed rats seem to be unlikely, as no such phenomenon has been described in the literature. Importantly, the number of footslips in BW was comparable between groups. Interestingly, the anxiety phenotype in LPS-affected adults was not apparent in EPM (Figure 6c,d). It is possible that the sensitivity of EPM was reduced due to previous experience with other tests (see also [63]). This discrepancy can be also explained by distinct subtypes of anxiety investigated by OF and EPM [64]. Results in anxiety-like behavior are inconsistent among other studies using similar models: some reported increased anxiety [45,65,66], while the others did not [45,52,67]. In any case, anxiety belongs to the symptomatology of both schizophrenia and ASD [68,69], although its expressions are very variable.

Interestingly, the offspring of LPS-treated mothers showed hypolocomotion in OF (Figure 6b) and decreased activity in EPM (Figure 6e). Animal models of schizophrenia usually exhibit hyperlocomotion, which is considered to be a model of positive symptoms based on common mesolimbic hyperactivity [70,71,72,73,74]. However, the situation in models prepared by LPS-induced MIA is not fully clear. Basta-Kaim et al., using a protocol comparable to the present study, observed enhanced locomotor activity in adult males and females [28] in contrast to our findings. Other studies reported all three possibilities: increased locomotor activity [45,75], no effect [67], or hypolocomotion [33,76], but using different protocols of LPS administration. Possibly, anxiety-like behavior leading to suppressed exploration was more prominent in some studies than the others. Another explanation could be a decreased motivation to explore the unknown environment. Amotivation and apathy (avolition) are serious negative symptoms of schizophrenia [77]. Unfortunately, we cannot decide between the hypotheses with the available data. ASD, unlike schizophrenia, is not linked with any specific changes in locomotor activity.

Our adult female, but not male LPS offspring also showed PPI impairment (Figure 6g). On the other hand, male LPS offspring showed a more prominent increase of acoustic startle response than LPS females in comparison to controls (Figure 6f). We did not measure startle response in adolescent rats, as the stressful experience could adversely affect subsequent behavioral tests. Other studies using prenatal LPS administration reported an increased startle response with normal PPI in adolescence [75,78]. In adulthood, elevated startle response, as well as PPI deficit, were described in the comparable model in both sexes [28,44]. Studies with other LPS models also found PPI deficits in adult rats and adolescents, but not before P30 [38,45,79]. PPI deficit belongs to the typical symptoms of schizophrenia and is believed to have good translational validity [80,81]. The deficit in sensorimotor gating has been also described in other neuropsychiatric disorders [82], including ASD [83,84], but less consistently. Hyper-reactivity to sensory inputs, including sound [6], as well as increased startle response without altered PPI [85], has been described in ASD patients.

ASD and schizophrenia are neurodevelopmental disorders [10] with both hereditary and environmental factors involved in their etiology. Prenatal infection represents an important risk factor with more or less serious consequences for the brain and behavior. In some cases, the indirect effect of the infectious agent acting though MIA might serve as a trigger, causing subtle changes that manifest only in susceptible individuals in later life [86]. It seems that the timing and chronicity of MIA play a crucial role in the final outcome, which could also partially explain the discrepancies between studies in animal models. Meyer et al. suggest that the development of schizophrenia or ASD is decided by the nature of the inflammation (acute, chronic, latent) in combination with specific genetic risk factors, leading to shared or unique symptoms [86]. Our results show that the sex of the offspring is an additional modulator of their future behavioral phenotype and also possible affinity to one or the other disorder. This is supported by sex differences observed in other studies discussed previously [33,38,44,45,66,78] and also by the influence of sex on prevalence, symptomatology, and severity of neurodevelopmental disorders in humans [41,42].

## 4. Materials and Methods

### 4.1. Animals

In total, thirteen Wistar rat dams (*n* = 7 experimental, *n* = 6 control; 230–330 g, 3–5 months old) were mated with six males (Velaz, Ltd., Prague, Czech Republic) to deliver 92 pups, which were used in our experiments. The offspring were divided into four groups (see Table 1). No more than three pups of each sex from the same litter were kept in the same group to ensure a balanced design.

The animals were bred at the National Institute of Mental Health (NIMH-CZ) in Klecany, Czech Republic, in standard Plexiglas boxes (44 × 28 × 23 cm) in an air-conditioned room with a 12 h/12 h light/dark cycle and food and water *ad libitum*, in groups of 2 or 3 individuals; only pregnant females were separated several days before delivery. All procedures took place during the light phase of the daily cycle. All experiments were approved by the Institutional Animal Care and Use Committee (Project of Experiment No. 23/2017) and complied with the Animal Protection Act of the Czech Republic, EU directive (2010/63/EU).

### 4.2. Model Preparation

After 5-day handling of the breeding rat females, we determined the estrous cycle using the protocol by Marcondes et al. ([87], see Appendix E). Then, females in the estrus phase, or proestrus–estrus interphase (Figure 7a) were mated with a male for 24 h. Pregnant rat females received 6 subcutaneous (s.c.) injections of bacterial lipopolysaccharide (LPS; Figure 7b) from *Escherichia coli* (L-3755, Serotype 026:B6; Sigma-Aldrich, Prague, Czech Republic) at a dose of 1 mg/kg and injection volume of 1 mL/kg, which was administered every other day from the seventh day of pregnancy to delivery, according to [28]. LPS was dissolved in 0.9% saline solution, the vehicle alone served as a control treatment. See the design of the whole experiment in Figure 7.

The pregnancy and delivery success was not lowered by the repeated administration of LPS (see the Appendix B and Figure A1 for more information). The pups were sexed at P6–8 and then left undisturbed until weaning at P28. The body weight of the offspring was measured before brain harvesting or before each set of behavioral testing. Three-day handling of adolescents preceded the behavioral examination at P45.

### 4.3. Brain Analysis

#### 4.3.1. Brain Tissue Harvesting

Rats (P28) were deeply anesthetized by xylazine (30 mg/kg) and ketamine (200 mg/kg) and transcardially perfused with 0.9% saline solution followed by 4% paraformaldehyde in 0.1M phosphate buffer at pH = 7.4. The brains were removed and post-fixed overnight in 4% paraformaldehyde in 0.1M phosphate buffer saline (PBS) at 4 °C and cryoprotected in 30% sucrose until they sank. Brains were cut on a cryostat (Leica CM1860 UV) into 50 µm thick coronal sections (1 in 5 series). For histology and immunohistochemistry, all brain sections were mounted on gelatin-coated slides.

#### 4.3.2. Toluidin Blue Staining and Brain Morphology Analysis

For morphological analysis, sections were stained by Toluidine blue (Nissl; Sigma-Aldrich, St. Louis, MO, USA). Measurements of lateral ventricles and dorsal hippocampus were done manually in ImageJ software (ImageJ, NIH, Madison, WI, USA) by a blinded researcher.

Cortical thickness was measured at approximately −4.56 mm from Bregma [88] in the first section where both dorsal and ventral hippocampal formation were present [89]. The sections were visualized under a light microscope Zeiss AxioImager Z1. Photographs were taken under a 2.5× objective (NA 0.06). Cortical thickness was analyzed using AxioVision Rel 4.8 software (Zeiss) as the distance between the boundary of the beginning of layer II and the beginning of the white matter beneath layer VI [89]. The average of the three measurements per one brain was analyzed separately for each hemisphere.

The size of lateral ventricles and size of the dorsal hippocampus was measured at approximately +0.00 mm and −2.28 mm from Bregma, respectively [88]. Tile scans were taken under 5× objective (NA 0.15, 512 × 512 pixels) with a confocal microscope Leica TSC SP8X. Ventricles were analyzed on one, while hippocampi were analyzed on three sections per rat. Areas of the structures were expressed as the percentage of total brain area [90].

#### 4.3.3. Immunohistochemistry and Analysis of PV+ Interneurons and Microglia

For immunofluorescence, free-floating brain sections were incubated with primary antibodies diluted in 0.1M PBS containing 0.3% Triton-X 100 overnight at 4 °C. The following primary antibodies were used: anti-Parvalbumin (PV, mouse monoclonal, 1:3000, Sigma-Aldrich, #P3088) and anti-Iba1 (rabbit polyclonal, 1:3000, Wako, #019-19741). After incubation with primary antibodies, sections were rinsed in 0.1M PBS and incubated with secondary antibodies in 0.1M PBS containing 0.3% Triton-X 100 for 2 h at room temperature. The following secondary antibodies conjugated with fluorophores were used: Donkey anti-rabbit AF488 (#711-545-152), Donkey anti-mouse AF488 (#715-545-150), 1:500, Jackson ImmunoResearch. After incubation with secondary antibodies, sections were rinsed in 0.1M PBS and coverslipped in ProLong Gold Antifade Reagent with DAPI (Cell Signaling Technology). Fixed immunofluorescence samples were viewed with a Zeiss Axio Imager Z1 fluorescent microscope (Carl Zeiss, Jena, Germany) or Leica TCS SP8X confocal system (Leica Microsystems Mannheim, Germany).

The number of PV+ cells. One brain section (right and left hemisphere separately) in the frontal cortex (Bregma +3.72 mm) and two brain sections (50 µm apart) in the dorsal hippocampus (Bregma −2.28 mm), from each animal, were used for PV+ cell counting. In the frontal cortex, two areas were evaluated: cingulate/secondary motor cortex (area 1) and prelimbic cortex (area 2). Frontal cortex sections were captured at 10× magnification (NA 0.4 CS) using a Leica TSC SP8X. We used z-stack images (1024 × 1024 pixels) with 1× digital zoom at a step size of 1.5 µm. In the dorsal hippocampus, the number of PV+ cells was measured in two regions: dentate gyrus (DG) and CA1–3 region, magnification 20× (NA 0.5, Zeiss). Immunopositive neurons were counted by using ImageJ in the complete subfield of each region.

The number of Iba1+ microglial cells in DG and hilus of the dorsal hippocampus (Bregma −2.92 mm) were measured. Three brain sections were analyzed per rat/per region. Images were acquired through objective 20× (0.75CS IMM CORR CS2, 1024 × 1024 pixels) by using a Leica TSC SP8X. All images were analyzed using the same threshold settings. Immunopositive microglial cells were counted manually in a complete sub-field of the region.

### 4.4. Behavioral Testing

The rat offspring of both sexes were tested repeatedly in a battery of behavioral tasks in adolescence (P45) and again in adulthood (P90). During an experimental day, all males were tested first, followed by all females; LPS and control animals were randomized. The apparatuses were always cleaned with water between individual sessions, or with a disinfectant between sexes and at the end of each experimental day, and dried by paper cloths. No other rats or observers were present in the experimental room during the session, except for the beam walking test. The behavioral tasks were performed in the same order as they are reported. Apart from these, we also trained the adult animals in the active place avoidance task [91,92] in the Carousel maze, testing spatial memory and cognitive coordination (reviewed in [93]), which was carried out at the Institute of Physiology CAS. However, even controls were unable to master the task, so it was not possible to make any conclusions (not shown here). After each day of experiments (to avoid stress before testing), the estrous cycle phase was determined in adult females.

#### 4.4.1. Open Field Test (OF)

The open field apparatus, consisting of a dimly illuminated (11 lux) square chipboard arena (70 × 70 cm) with black-laminated floor and opaque walls (40 cm high), serves as a test of spontaneous locomotor activity and anxiety. The animal was put into the center of the arena and recorded by an overhead camera for 10 min. Recorded videos were analyzed offline automatically using Viewer 3 software (Biobserve, Bonn, Germany), which tracked the position of the rat. The total walked distance and time spent in the center of the arena (50 × 50 cm) were evaluated.

#### 4.4.2. Beam Walking Test (BW)

Primarily, the task assesses sensorimotor coordination. The animal had to walk along a 2 m long wooden beam, at first wide (4.5 cm) and then narrow (2.2 cm), located 1 m above the floor, to reach its homecage at the opposite end. During the training period, the rat was prompted to walk only 0.5 m or 1 m of the beam length before the full length in the testing phase (see the order of trials in Table 2).

The performance was observed by two experimenters, watching the animal from opposite sides. The time needed to traverse the beam and the number of footslips (when at least one of the rat’s paws slipped from the upper surface of the beam) were evaluated.

#### 4.4.3. Elevated Plus Maze Test (EPM)

The plus maze tests anxiety and spontaneous exploration. The cross-shaped black apparatus elevated 50 cm above the floor consisted of two closed arms (safer for the animals) and two opposite open arms (each arm 50 × 10 cm, closed arms had walls 30 cm high). Illumination was 100 and 630 lux in the closed and open arms, respectively. Each rat was placed into the center facing an open arm and then explored the maze for 5 min.

Rat behavior was recorded by an overhead camera, and the videos were analyzed manually in BORIS software [94]. Overall activity (total arm visits), the proportion of open arm visits to total arm visits, and frequency of risk assessment behavior (looking out from the closed arms without leaving it) were assessed.

#### 4.4.4. Social Interactions

Social behavior was measured by a simple social interaction test in a neutral environment. In our case, two animals of the same sex from the same group but different cages were placed into the OF arena already familiar to them, dimly illuminated (11 lux), facing the opposite corners, and were left undisturbed and recorded by an overhead camera.

Animal behavior (5 min of the session) was analyzed manually by a blinded observer using BORIS software [94]; each animal from the pair was scored separately. The main scored categories were duration of anogenital exploration and duration of non-anogenital social contact (sniffing or touching the other rat’s body except the anogenital area or tail). Other scored parameters (following the social partner, evade, crawling, play or fight; self-grooming, freezing) were too rare for statistical evaluation, or did not differ between groups and are not reported.

#### 4.4.5. Ultrasonic Vocalization Recording (USV)

During each social interaction session, ultrasonic vocalization as an indicator of social communication between rats was recorded by an Ultramic 250 K microphone (Dodotronic, Italy) at a sampling frequency of 250 kHz.

Audacity 2.1.2. software (Pittsburgh, PA, USA) was used for both acquisition and analysis, which was done by a blinded observer who manually marked individual calls. Each pair of rats was taken as a single unit, since the calls could not be attributed to individual rats.

Firstly, individual elements of high-frequency (above 50 kHz) vocalization were marked. Those were further subdivided into trill-like (with rapid periodic oscillations over a wide range of frequencies, and regular wave-shaped calls), flat (vocalizations with uniform frequency lasting more than 50 ms), and other (not fitting any other category). Simple calls (consisting of a single element) and composite calls (consisting of multiple elements) were distinguished. The total number of calls, the average duration of a simple call, the average duration of a composite call, and the proportion of simple versus composite calls (calculated as a percent contribution of simple calls to the total time spent by vocalization) were used for further analysis, as well as the number of trill-like elements. Flat elements were too rare for meaningful statistics. Examples of vocalizations are shown in Figure 4f.

#### 4.4.6. Acoustic Startle Reaction and Prepulse Inhibition (PPI)

In this task, the animal was closed in a tight tube placed in a soundproofed box (SR-LAB Startle Response System, San Diego Instruments, San Diego, CA, USA) which measured the startle jumping reflex of the animal as a reaction to strong acoustic stimuli. Firstly, the animal stayed in the apparatus for 5 min for habituation and listened only to white noise (70 dB). Then, the rat was exposed to 50 trials with acoustic stimuli in a pseudorandom order (alternated phases were no stimulation and acoustic pulse 120 dB with or without prepulse) and irregular inter-trial intervals (12–20 s). Three different intensities of prepulses were used: 3, 5, or 10 dB above the white noise level (i.e., 73, 75, and 80 dB).

The degree of startle reaction to the strong acoustic stimulus (120 dB without a prepulse) was measured separately. Prepulse inhibition is a measure of sensorimotor gating when a shorter and weaker prepulse preceding the main pulse physiologically inhibits the startle reaction. PPI (shown in %) was calculated according to the formula: [PPI = 100 − (PP/P120) × 100]
where the startle reaction after the acoustic pulse preceded by a prepulse (PP) was divided by the startle reaction after the acoustic pulse without a prepulse (P120).

Only adult rats were tested here because of the stressful nature of the task. The number of animals used for statistical analysis was lower in this task because some data were lost due to a technical error.

### 4.5. Statistical Analysis

All the data were statistically analyzed in the IBM SPSS Statistics software. Three-way analysis of variance with repeated measures (3wANOVA-RM) was used for most of the behavioral parameters, and four sections of brain area measurement, with group (LPS or saline) and sex (males or females) as between-subject factors and age (P45 or P90) or section (1, 2, 3 or 4), respectively, as a within-subject factor. In case there was a significant group–age or sex–age interaction, we used simple effects to specify the nature of the interaction and to identify the age where the groups or sexes differed. Two-way ANOVA (2wANOVA) with the same between-subject factors was used for the rest of the brain parameters, and also for several behavioral parameters, where P45 and P90 were analyzed separately. In PPI averaged for all prepulses, the *t*-test was used to show a significant difference specifically in females. The data was log-transformed to meet parametric assumptions in case of non-normal distribution. In BW number of footslips, the transformation did not lead to normalization of the data, and so the negative binomial model with log estimates values was applied. Significance was accepted at *p* ≤ 0.05. Adult females in the estrous phase at the day of testing were excluded from the analysis because estrus affects behavior [95,96]. For the numbers of animals used in each test, see Appendix A.

## 5. Conclusions

Our current findings revealed that chronic MIA, elicited by the administration of bacterial LPS to pregnant rat dams, affected brain development (macrocephaly, decreased number of PV+ interneurons in the frontal cortex) and behavior (social and communication deficits, anxiety, hypoactivity, increased startle reaction, PPI deficit) of the offspring. Importantly, while changes in the brain of weaned (P28) rats were similar in both sexes, behavioral deficits at P45 and P90 differed between males and females, indicating that the behavioral outcomes developing later in life are strongly modulated by sex. This points to the possibility that a comparable prenatal insult may lead to one or the other disorder depending on additional factors, including sex. Our results demonstrate that the common practice of omitting females from preclinical research can lead to misleading or incomplete conclusions about model validity.

## Figures and Tables

**Figure 1 ijms-22-03274-f001:**
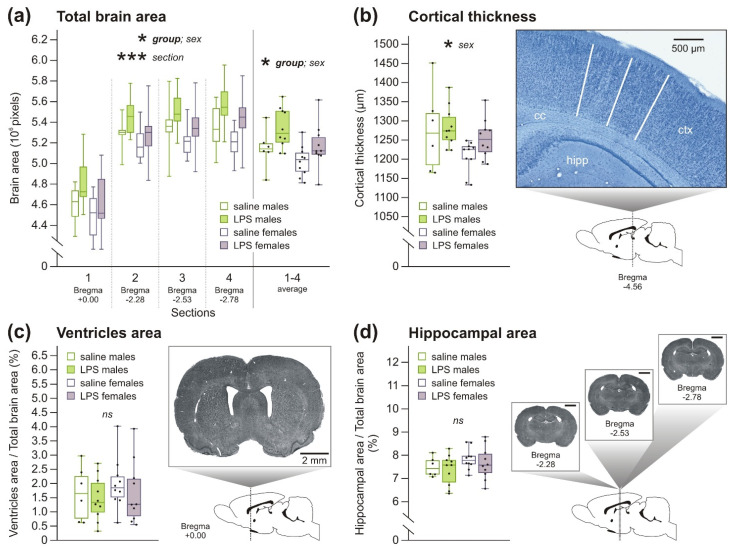
Brain morphology in P28-old rats. (**a**) Areas of four different brain sections were larger in both males and females of lipopolysaccharide (LPS)-treated dams compared to controls, as three-way analysis of variance with repeated measures (3wANOVA-RM) confirmed. In addition, males had larger brain area than females. (**b**) Cortical thickness in the LPS-exposed offspring was not significantly different from the control offspring, as it was shown by the two-way ANOVA (2wANOVA), but there was the sex effect, as males had thicker cortex than females. (**c**,**d**) Ventricles area (**c**) and dorsal hippocampal area (**d**) shown as a percentage of whole brain area, both analyzed by 2wANOVA, were not significantly affected by prenatal exposure to LPS. Scale bars show 2 mm. In the graphs in panels (**b**–**d**), each boxplot shows an average value for the left and right hemispheres. The boxplots show median, first and third quartile and minimum and maximum values; the dots show individual values. * *p* < 0.05, *** *p* < 0.001, ns = no significance. cc = corpus callosum; ctx = cortex; hipp = hippocampus.

**Figure 2 ijms-22-03274-f002:**
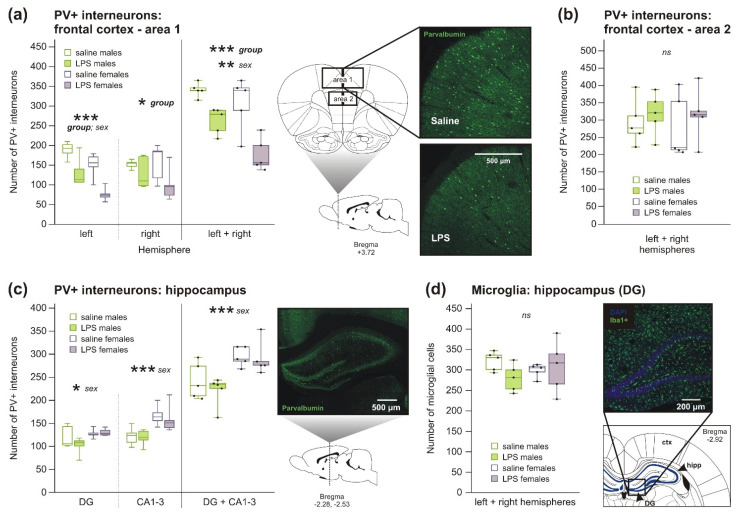
Parvalbumin-positive (PV+) interneurons and microglia in P28-old rats. (**a**) Number of PV+ interneurons was decreased in the upper part of the frontal cortex of both hemispheres (area 1), with the higher difference found in the left hemisphere, in both males and females prenatally exposed to LPS, as 2wANOVA analyses showed. Moreover, there was a sex difference with males showing higher density of PV+ interneurons, specifically in the left hemisphere. The graph shows averaged values from left and right hemispheres separately, and for an average of both. (**b**,**c**) The difference between groups in the number of PV+ interneurons located in lower part of the frontal cortex (area 2) (**b**), as well as in the dorsal hippocampus (**c**) was not significant, according to 2wANOVA. However, males in general showed lower number of PV+ interneurons in the dorsal hippocampus compared to females, more profound in CA1-3 areas than the dentate gyrus. (**d**) The number of microglia in the dentate gyrus of the dorsal hippocampus was also not affected by prenatal experience with LPS, according to 2wANOVA. The boxplots show median, first and third quartile, and minimum and maximum values; the dots show individual values. * *p* < 0.05, ** *p* < 0.01, *** *p* < 0.001, ns = no significance. ctx = cortex; DG = dentate gyrus; hipp = hippocampus.

**Figure 3 ijms-22-03274-f003:**
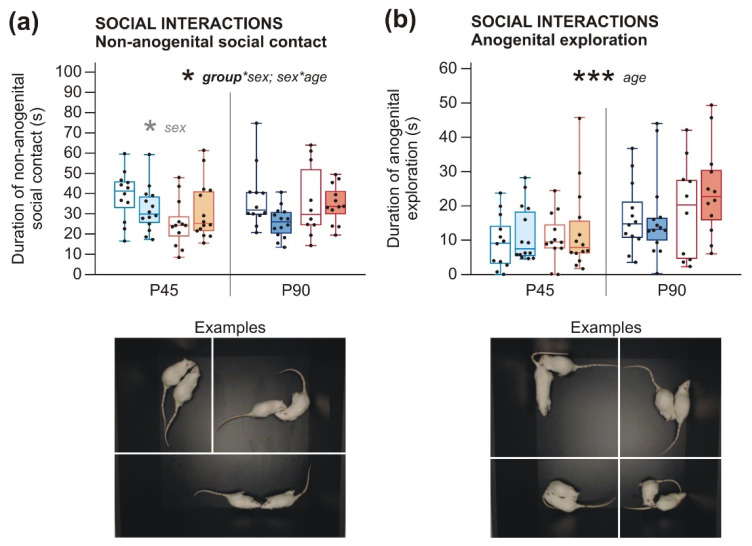
Social behavior in P45 and P90 rats. (**a**) Duration of non-anogenital social contact was lower in males of LPS-treated dams than males of control dams at P45 as well as P90, as shown by 3wANOVA-RM. However, LPS females differed from control females neither at P45 nor at P90. In addition, sex difference measured by 2wANOVA for P45 showed that males participated in non-anogenital social contacts for longer time than females in adolescence, but not in adulthood. (**b**) Duration of anogenital exploration was not shown to be affected by LPS. Adult animals spent more time by anogenital exploration than adolescents, as shown by 3wANOVA-RM. The boxplots show median, first and third quartile, and minimum and maximum values; the dots show individual values. * *p* < 0.05, *** *p* < 0.001.

**Figure 4 ijms-22-03274-f004:**
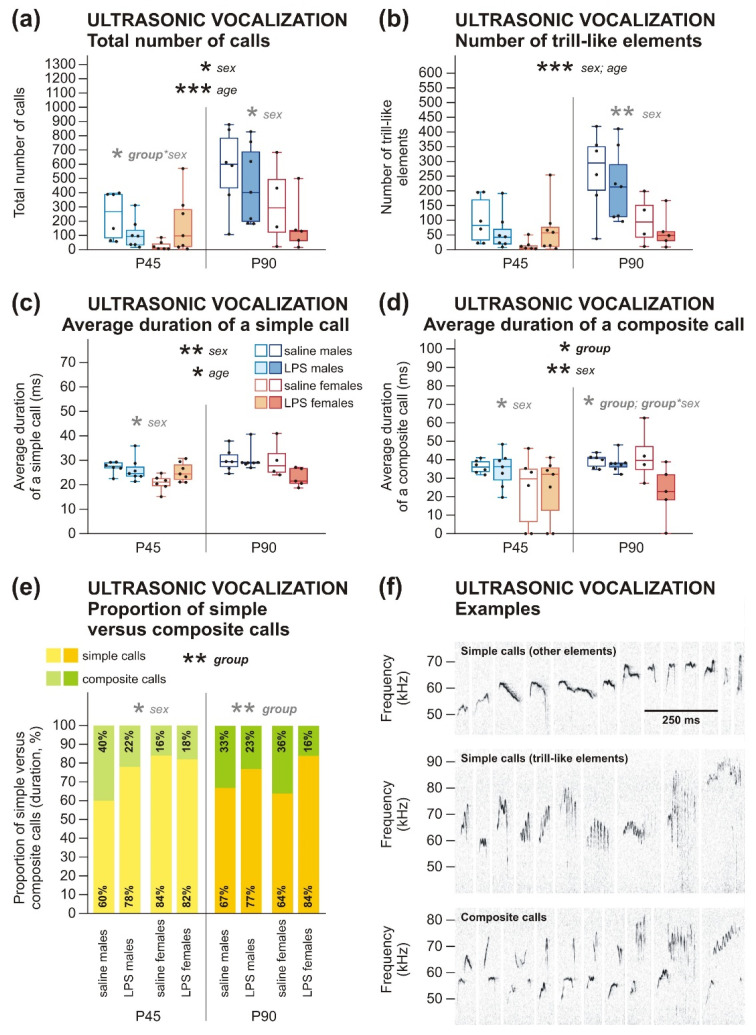
Ultrasonic vocalization (USV) in P45 and P90 rats. (**a**) In adolescence, a distinct effect of LPS on males and females was found in total number of calls. 2wANOVA revealed that LPS males emitted less vocalizations compared to control males, but LPS females emitted a higher amount of vocalization compared to control females. In adulthood, the difference was not present. In addition, males emitted a higher total number of calls than females. Adult animals produced much more vocalizations than adolescents. (**b**) Trill-like USV was not affected by LPS in any age. However, males emitted a higher number of trill-like elements than females, and also adult rats had a higher number of trill-like elements in comparison to adolescents. (**c**,**d**) Average duration of a simple call did not differ between groups. However, adult females prenatally exposed to LPS showed a shorter average duration of a composite call than control females, as 2wANOVA revealed. 3wANOVA-RM also revealed longer average duration of simple calls and composite calls in males in comparison to females. Average duration of simple calls was significantly higher in adult animals than adolescents; in the composite calls, the same tendency was present as a strong trend. (**e**) The proportion of simple versus composite calls was changed in LPS-exposed rats, as shown by 3wANOVA-RM and 2wANOVA. LPS-exposed rats of both sexes, especially at P90, spent shorter time by emission of composite calls than simple calls, compared to controls. In addition, P45 males spent a longer time emitting the composite calls, relative to the simple calls than P45 females. (**f**) Examples of analyzed types of USV: simple calls, trill-like elements, composite calls. The boxplots show median, first and third quartile, and minimum and maximum values; the dots show individual values. * *p* < 0.05, ** *p* < 0.01, *** *p* < 0.001.

**Figure 5 ijms-22-03274-f005:**
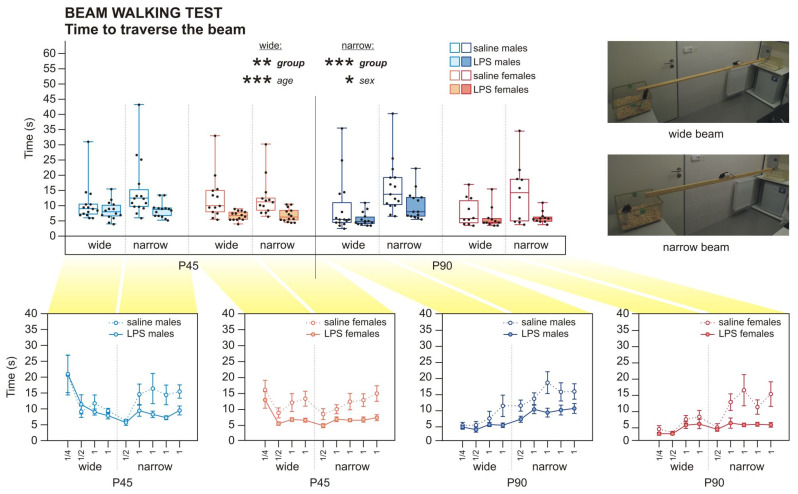
The beam walking test in P45 and P90 rats. LPS-exposed rats of both sexes spent less time by crossing the wide or narrow beam compared to controls at both ages, according to 3wANOVA-RM. Males took more time to cross the narrow beam than females. Adult rats spent less time traversing the wide beam than adolescent rats. The upper panel show data averaged only for whole-beam crossings. The boxplots show median, first and third quartile, and minimum and maximum values; the dots show individual values. In the x axes of the lower panels, the numbers “1/4” and ”1/2” mean training phase (0.5 or 1 m, respectively, of the beam) and ”1” means the test phase (the whole 2-m long beam). The values shown in the lower panels indicate means ± S.E.M. * *p* < 0.05, ** *p* < 0.01, *** *p* < 0.001.

**Figure 6 ijms-22-03274-f006:**
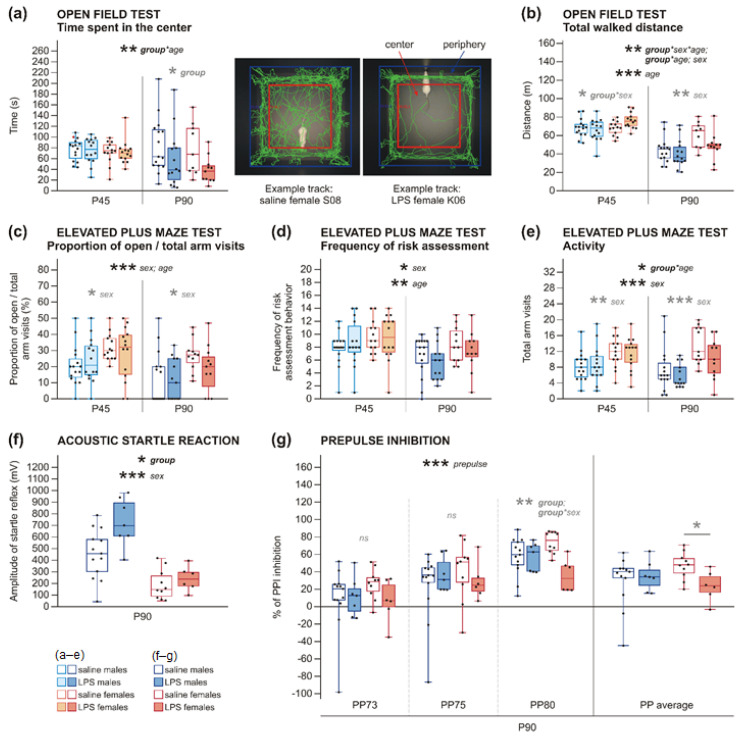
Non-social behavior in P45 and P90 rats. (**a**) In the open field test (OF), the rats of both sexes prenatally exposed to LPS spent less time in the center of the arena in comparison to saline-exposed rats, as revealed by 3wANOVA-RM with simple effects and 2wANOVA, but only in adulthood, not in adolescence. (**b**) Total walked distance measured in OF was changed differently at P45 and P90 by prenatal exposure to LPS. In adolescence, LPS males did not differ from control males; in contrast, LPS females walked a longer distance than control females. In adulthood, both LPS males and females tended to walk a shorter distance than the control animals. In addition, males showed decreased locomotor activity compared to females as well as adults compared to adolescents. The results were evaluated by 3wANOVA-RM and 2wANOVA. (**c**,**d**) In the elevated plus maze test (EPM), the proportion of open arm visits to total arm visits (**c**) and frequency of risk assessment behavior (**d**) were not significantly affected by LPS administration to pregnant dams. However, sex differences were found by 3wANOVA-RM: males had lower open arm visits ratio and also lower risk assessment behavior compared to females. Similar result was found in adult rats in comparison to adolescents. (**e**) Activity in EPM was not changed by LPS experience at P45, but it slightly decreased at P90 in LPS males as well as females, as shown by 3wANOVA-RM and as a trend by 2wANOVA. In males, the activity was also decreased in comparison to females. (**f**) LPS-exposed animals showed a higher startle response to strong acoustic stimuli (120 dB), compared to controls, according to 2wANOVA. The difference was more manifested in males. Moreover, the analysis revealed a higher startle response in males compared to females. (**g**) Prepulse inhibition deficit was found by 2wANOVA and *t*-tests in LPS females, especially after strong stimulus with 80-dB prepulse (PP), but not in LPS males. The boxplots show median, first and third quartile, and minimum and maximum values; the dots show individual values. * *p* < 0.05, ** *p* < 0.01, *** *p* < 0.001.

**Figure 7 ijms-22-03274-f007:**
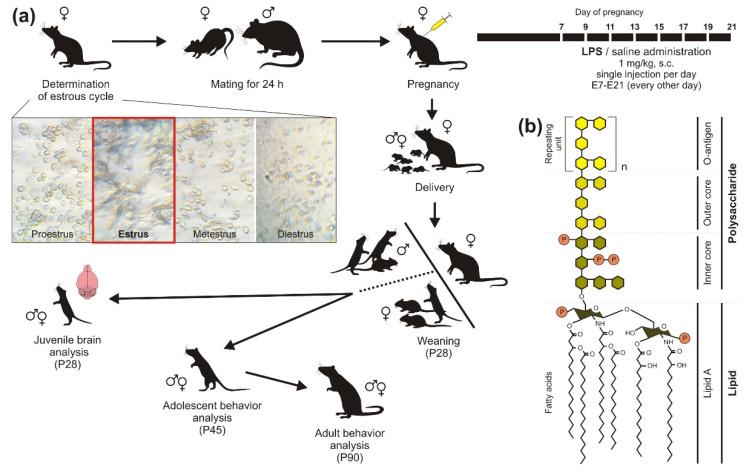
(**a**) The design of the experiment. In rat females, estrous cycle was determined for approximately one week. Estral females were mated with a male for 24 h. During pregnancy, females received six subcutaneous (s.c.) injections of lipopolysaccharide (LPS) at a dose of 1 mg/kg, or 0.9% saline solution (control), from embryonic day E7 to E21 (delivery), every other day. The offspring of both sexes were used for brain analysis at postnatal day P28 (weaning) or for the behavioral testing in adolescence (P45), which was repeated in adulthood (P90). (**b**) The structure of a bacterial LPS; according to [36].

**Table 1 ijms-22-03274-t001:** Group specification and the number of animals per group.

Group	Number of Animals and Litters Per Group (*n*)		
	Brain Analysis (P28)		Behavior Analysis (P45, P90)
	Brain Morphology	Immunohistochemistry	
saline males	*n* = 6 (4 litters)	*n* = 5 (4 litters)	*n* = 15 (5 litters)
LPS males	*n* = 10 (7 litters)	*n* = 5 (5 litters)	*n* = 14 (7 litters)
saline females	*n* = 10 (4 litters)	*n* = 5 (4 litters)	*n* = 13 (5 litters)
LPS females	*n* = 10 (7 litters)	*n* = 5 (5 litters)	*n* = 14 (7 litters)

LPS = lipopolysaccharide; P = postnatal day.

**Table 2 ijms-22-03274-t002:** Design of the beam walking test.

	Training		Test		Training	Test			
**Trial**	**1**	**2**	**3**	**4**	**5**	**6**	**7**	**8**	**9**
**Width**	4.5 cm	4.5 cm	4.5 cm	4.5 cm	2.2 cm	2.2 cm	2.2 cm	2.2 cm	2.2 cm
**Length**	0.5 m	1 m	2 m	2 m	1 m	2 m	2 m	2 m	2 m

## Data Availability

The data presented in this study are available upon request from the corresponding authors.

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
