# Peer review of "Behavioral Alterations and Decreased Number of Parvalbumin-Positive Interneurons in Wistar Rats after Maternal Immune Activation by Lipopolysaccharide: Sex Matters"

_ijms, 2021, doi:10.3390/ijms22063274_

Round 1
Reviewer 1 Report
I have gone through the whole manuscript. Article entitled “Behavioral alterations and decreased number of parvalbumin-positive interneurons in Wistar rats after maternal immune activation by lipopolysaccharide: sex matters.”, concept is appealing and well written. This article is presented for readers to follow the study rationale. The authors make a comprehensive contribution to neuro-developmental studies. It summarizes that gender of the offspring plays a crucial role in the development of the MIA-induced behavioral alterations, whereas changes in the brain in young animals are sex-independent but in older age brain development is sex dependent.
I have just few comments/suggestions:
- It is well known that LPS induce pro-inflammatory cytokines. It will be more supportive if Authors add cytokine levels in result section and compare to sex (Male and Female).
- Did this study provide any impact on therapeutic regimen?
Author Response
Point 1: It is well known that LPS induce pro-inflammatory cytokines. It will be more supportive if Authors add cytokine levels in result section and compare to sex (Male and Female).
Response 1: Thank you very much for your suggestion. We agree that a comparison between males and females in cytokine level (and maybe its correlation with behavioral findings) would be interesting. However, we hadn’t collected samples for cytokine levels measurements during the running project and now, as the project finished, it is not possible to add these analyses into the current manuscript. Nevertheless, it is a great question for our further project.
Point 2: Did this study provide any impact on therapeutic regimen?
Response 2: One has to bear in mind that our work concerned just a single risk factor in a rat model, a limited approximation of the situation in human patients, so we think it is too early for any therapy suggestions. This question would be more suitable for a comprehensive review. An obvious recommendation would be to prioritize prevention and treatment of infections during pregnancy, and perhaps also to monitor mental health of the individuals at high risk of schizophrenia. Once the causal links between infection and psychiatric disorders become established, it may open the way towards causal therapy/prevention, perhaps by modulating the immune response in the mother or the child, but clearly we are not yet there.
Reviewer 2 Report
The manuscript by Iveta Vojtechova et al, describes the “impacts of MIA on the brain and behavior of adolescent and adult offspring, as a rat model of these neurodevelopmental disorders.”
This work is very thorough and very well detailed and described. I must congratulate the authors on their work. I felt sometimes that the information in annex should be in the main text, because the broad interest of the work is indeed the characterization of the model in its many aspects, while for example the tables with the statistics could be put in an annex to allow the reads to better follow the results. Also, because the figures are very clear, also in terms of statistics, this would not create any lack of clarity to the readers.
My questions are minor, but in as much as it is possible for the authors to go back to their data, I would like to make a few points that, in this reviewer’s opinion, may strengthen the author’s interpretation and conclusions.
- histological and IHC analysis
For you histological and IHC analysis your methodology for assessing changes in macroscopic brain areas is mostly done in a reduced number of sections per animal. For example, the size of lateral ventricles is assessed in one section, (approximately +0.00 mm ) and size of the hippocampus using 3 sections all of which were from the dorsal hippocampus (at approximately -2.28 mm from Bregma). Why did you decide to restrict your analysis to these coordinates/areas?
There are many regional differences within the hippocampal formation, and the PV+ distribution and changes across both age, namely adolescence, and schizophrenia. These changes are more noticeable in the ventral hippocampus, rather than the dorsal. It would add greater details to the study if: 1) information for PV+ in ventral HIP was also included, and 2) if analysis of the adult brains (from the animals that underwent behavioral testing), rather than P28 alone, was added.
It would be interesting to determine if the expected changes across age that are described for both pre-frontal cortex and ventral hippocampus are blunted in this model and offer some new insights into the GABAergic network alterations. (See for example Caballero et al, (2013) Region-specific upregulation of parvalbumin-, but not calretinin-positive cells in the ventral hippocampus during adolescence Hippocampus. 2013 Dec; 23(12): 1331–1336.)
Despite the fact, that the MIA model is a developmental model, that targets specifically gestation, its effects are long-lasting and so the early evaluating (P28) would be nicely complemented by the latter evaluation (P90) if these brain samples are available to do so.
Otherwise, please make sure you mention that the changes are only related to dorsal hippocampus in your results and discussion sections.
2.Behavior experiments:
In the tests done in the same animals at P45 and then P90, that depend on exploratory behavior of novel environments, I think the cautionary note in the annex, should be in the main text, that in factor contributing to the “age” differences may very well be just lack of novelty.
For example, if the authors analyze just the first 5 min of the open field exploratory behavior at both ages (P45 and P90), would they still find the differences ascribed to age in distance traveled.
In fact, the authors acknowledge this in the discussion of the “lack of EPM changes” ascribed to age “It is possible that the sensitivity of EPM was reduced due to previ[1]465 ous experience with other tests (see also [62])”. Lack of novelty in the EPM is especially noteworthy since the lighting intensity used for this experiment produced an aversive environment, that could explain both the lack of visits of a significant portion of P90 males and a reduced total arm visit. Have the authors been able to use these experimental settings (100 and 630 lux in closed and open arms respectively) to detect anxious behavior previously?
The authors have done an excellent job in their description of the experimental settings and conditions. I would only suggest that in figure 5 they may introduce in the legend that ¼ and ½ correspond to training sessions with the respective fractions of the total beam walk, for clarity of the interpretation of the data.
